# Redox Mechanism of Azathioprine and Its Interaction with DNA

**DOI:** 10.3390/ijms22136805

**Published:** 2021-06-24

**Authors:** Mihaela-Cristina Bunea, Victor-Constantin Diculescu, Monica Enculescu, Horia Iovu, Teodor Adrian Enache

**Affiliations:** 1Laboratory of Multifunctional Materials and Structures, National Institute of Materials Physics, Atomistilor 405A, 077125 Magurele, Romania; mihaela.bunea@infim.ro (M.-C.B.); victor.diculescu@infim.ro (V.-C.D.); mdatcu@infim.ro (M.E.); 2Advanced Polymer Materials Group, University Politehnica of Bucharest, 1-7 Gh. Polizu, 011061 Bucharest, Romania; iovu@tsocm.ro

**Keywords:** azathioprine, redox mechanism, DNA biosensor, DNA interaction, mass spectrometry

## Abstract

The electrochemical behavior and the interaction of the immunosuppressive drug azathioprine (AZA) with deoxyribonucleic acid (DNA) were investigated using voltammetric techniques, mass spectrometry (MS), and scanning electron microscopy (SEM). The redox mechanism of AZA on glassy carbon (GC) was investigated using cyclic and differential pulse (DP) voltammetry. It was proven that the electroactive center of AZA is the nitro group and its reduction mechanism is a diffusion-controlled process, which occurs in consecutive steps with formation of electroactive products and involves the transfer of electrons and protons. A redox mechanism was proposed and the interaction of AZA with DNA was also investigated. Morphological characterization of the DNA film on the electrode surface before and after interaction with AZA was performed using scanning electron microscopy. An electrochemical DNA biosensor was employed to study the interactions between AZA and DNA with different concentrations, incubation times, and applied potential values. It was shown that the reduction of AZA molecules bound to the DNA layer induces structural changes of the DNA double strands and oxidative damage, which were recognized through the occurrence of the 8-oxo-deoxyguanosine oxidation peak. Mass spectrometry investigation of the DNA film before and after interaction with AZA also demonstrated the formation of AZA adducts with purine bases.

## 1. Introduction

The medical use of azathioprine (6-([1-methyl-4-nitro-1Himidazol-5-yl]-sulphonyl)-7H-purine, AZA), which metabolically breaks down to 6-mercaptopurine in the human body, dates back to 1963 [1,2]. Being a purine analog, AZA blocks purine metabolism and DNA synthesis, and in recent times has been administrated as an immunosuppressive (mainly to prevent renal transplant rejection) and antileukemic drug to treat autoimmune diseases such as ulcerative colitis, rheumatoid arthritis, dermatitis, lupus erythematosus, and Crohn’s disease [3,4,5]; however, there have been many reports about the high risk of skin cancer for patients exposed to solar radiation under AZA treatment, a fact that limits the administration of AZA [6,7,8,9,10]. In order to decrease this side effects, appropriate experimental models should be investigated at the laboratory level, and after the underlying principles are understood, integrated with pharmaceutical studies. Generally, the biological processes are governed by the electron transfer reactions, which means that the electrochemical methods can successfully be used to simulate complex in vivo biochemical interactions. In order to sustain the pharmacokinetics studies, one of the aims of this study was to investigate the electrochemical behavior and electron transfer mechanism of AZA, using cyclic and differential pulse voltammetry on a glassy carbon electrode.

To date, electrochemical studies have been performed investigating the voltammetric behavior of AZA, but a complete redox mechanism has not been proposed to date [11,12,13]. Several scientific articles have shown the improvement of azathioprine detection limits using various modified sensing platforms based on graphene, which can be decorated with Ag nanoparticles, incorporated into natural polymers such as chitosan, or functionalized with ionic liquids [12,14,15,16]. Glassy carbon electrodes are another type of surface that are widely used for the development of new sensors with nanodiamond–graphite chitosan film or with carbon nanoparticle–nafion film in order to improve the sensitivity of the voltammetric determination of azathioprine [14,15].

On the other hand, DNA damage and its interaction with biologically active molecules are important fundamental issues in life science. The achievement of more efficient targeted pharmaceutics has been the subject of recent investigations, based on DNA-drug interaction [17,18,19]. The interaction of DNA with endobiotics or xenobiotics can take place via various mechanisms, which may be assisted by enzymatic reactions [20,21,22,23]. Among the chemical reactions that are not catalytically assisted, it is worth highlighting the electrostatic interactions and the intercalation of chemical molecules between DNA strands, which results in conformational modifications of the secondary structure of DNA [24,25].

Over the last decades, electrochemical biosensors have been proven to be reliable analytical tools for the elucidation of the interaction mechanisms of DNA with pharmaceutical compounds [26,27,28,29]. The advantages offered by electrochemical biosensors over other screening methods include the possibility for automation and real-time analysis, improved reproducibility and speed, and low production costs [27,30,31,32].

In this context, the goal of this study was the investigation of DNA–AZA interaction using electrochemical DNA biosensors, scanning electron microscopy, and mass spectrometry. The novelty of this study was the use of DNA electrochemical biosensors for the investigation of morphological DNA modifications upon interaction with AZA. Biosensors based on dsDNA, poly(A), and poly(G) were used to investigate the AZA–DNA interaction. The electrochemical characterization of the in situ interaction of AZA with DNA or purine bases on a glassy carbon modified electrode combined with mass spectrometry and scanning electron microscopy provided essential information about the effects of this drug on DNA.

## 2. Materials and Methods

### 2.1. Reagents and Solutions

Azathioprine (AZA), 1 methyl-4 nitro-imidazole (IMI), 6-mercaptopurine (Merc), double-strand DNA (dsDNA), polyadenylic acid (poly(A)), and polyguanylic acid potassium salt (poly(G)) were acquired from Sigma-Aldrich.

According to the supplier, dsDNA (catalogue number D1501) is extracted using a method that causes shearing, yielding a highly polymerized mixture of double- and single-stranded DNA. The double-stranded DNA is the predominant form. The base distribution for DNA from calf thymus samples is 41.9 mol% G-C and 58.1 mol% A-T. Poly(A) (catalogue number P9403) was prepared from adenosine diphosphate (ADP) with polynucleotide phosphorylase, while poly(G) (catalogue number P4404) was synthetically produced in *Thermus thermophilus* from guanosine diphosphate (GDP). All of the substances were used without further purification.

The electrolyte solutions of acetate buffer (pH = 4.5), phosphate buffer (pH = 7), and ammonia buffer (pH = 9) were prepared with analytical-grade reagents and purified water from a Milli-Q system (conductivity of less than 0.1μS cm^−1^).

The stock solution of AZA was prepared daily in DMSO and kept at 4 °C and protected from light. Solutions of different concentrations of azathioprine were prepared by dilution of the appropriate quantity in the desired buffer. All experiments were carried out in a dark environment.

Stock solutions of 35 mg mL^−1^ dsDNA and 3.5 mg mL^−1^ poly(A) and poly(G) were prepared in purified water.

All experiments were performed at room temperature (25 ± 1 °C).

### 2.2. Instrumentation

#### 2.2.1. Electrochemical Measurements

The measurements were performed using a computer-controlled Ivium potentiostat with IviumSoft version 2.219 (Ivium Technologies, Eindhoven, The Netherlands). Measurements were carried out using a glassy carbon working electrode (*d* = 1.6 mm), a Pt wire counter, and an Ag/AgCl (3 M KCl) reference electrode in a single-compartment 2 mL electrochemical cell. Before each experiment, the GCE was polished using diamond spray (particle size 1 μm) on a microcloth pad, rinsed with Milli-Q water, and electrochemically pre-treated by recording various DP voltammograms in buffer-supporting electrolyte until a steady-state baseline voltammogram was obtained. All measurements were performed in N_2_-saturated solutions in the dark in order to prevent the UV degradation of the immunossupresive drug.

Cyclic voltammetry (CV) was performed for scan rates ranging from 10 to 500 mV s^−1^ with a 2 mV step potential from E_start_ = 0V, E_min_ = −1 V, and E_max_ = 1.4 V. Differential pulse (DP) voltammetry values were recorded with a pulse amplitude of 50 mV, pulse width of 100 ms, and scan rate of 5 mV s^−1^. All of the presented voltammograms were background-subtracted and baseline-corrected using the IVIUM soft program tools. This mathematical treatment reduces the peak heights by up to 10% and was used for the presentation of all experimental voltammograms to allow easier and clearer identification of the peaks. The values for peak currents presented in all graphs were determined from the original untreated voltammograms.

#### 2.2.2. Field-Emission Scanning Electron Microscopy (FESEM)

The morphology of the samples was investigated using a Gemini 500 Carl Zeiss field-emission scanning electron microscope (FESEM) working in both high-vacuum (HV) and variable-pressure (VP) modes from 0.2 to 30 kV, equipped with LaB6 filament, InLens and SE2 detectors, NanoVP mode, and a Bruker QUANTAX 200 energy-dispersive X-ray spectrometer (EDS) with an XFlash^®^6 silicon drift detector (SDD), an energy resolution <129 eV at Mn-Ka, and Peltier cooling. The above setup was used to investigate the elemental compositions of the samples.

#### 2.2.3. Mass Spectrometry

All mass spectra were acquired using an ESI ion source Bruker Daltonik amaZon speed ion trap mass spectrometer (Bruker Daltonik, Bremen, Germany). The ion source was operated in positive mode with nitrogen as the drying gas, with a flow rate of 5 L/min at 180 °C, nebulizer pressure of 7.5 psi, capillary voltage of 4.5 kV, and end plate offset of 0.5 kV. Optimum ion transfer was achieved by automatically running the system smart parameter setting in order to optimize the ion transfer for the desired *m*/*z* value. Charge control of the ion trap was activated with a target value of 200,000 and a maximum accumulation time of 10 ms. Samples of 1 mM azathioprine in methanole were injected directly into the mass spectrometer at a flow rate of 180 μL h^−1^. Mass spectra were recorded for *m*/*z* values between 50 and 380 at 5200 amu/s in maximum resolution scan mode and ten scans were averaged into one mass spectrum.

### 2.3. Biosensor Preparation and Incubation Procedure

The DNA biosensors were obtained by covering the GC surface with one 2 μL drop containing 35 mg mL^−1^ dsDNA gel and allowed to dry in air. For control experiments, poly(G) and poly(A) biosensors were prepared following the same protocol. The incubation procedure consisted of immersion of the biosensors for different periods of time in solutions with various concentrations of AZA in pH 4.5 0.1 M acetate buffer with or without an applied voltage (–0.600 V). After this procedure, the electrode was rinsed with purified water in order to remove the weakly bound or unbound molecules and measured in acetate buffer using the DPV method.

The acidic digestion of DNA before and after incubation with AZA involved removal of the DNA film from the electrode surface; thus, the DNA film weighing about 350 µg (10 μL drop containing 35 mg mL^−1^ dsDNA gel) was removed mechanically from the electrode surface and treated with HClO_4_ 9M over 10 min. The digestion was stopped with NaOH 9M after 10 min digestion. All digested DNA samples were diluted with purified water and subjected to MS spectrometry.

## 3. Results and Discussion

The objective of this investigation was to understand the interactions between AZA and DNA by using dsDNA electrochemical biosensors. The first part of this study was dedicated to the understanding of the electrochemical behavior of AZA through comparative studies with 1-methyle-4-nitro imidazole, using voltammetric techniques and a GC electrode. In the second part of the study, the DNA electrochemical biosensors were used in order to observe changes of the conformation of the DNA immobilized at the GC electrode surface. Different DNA-based biosensors, such as dsDNA, poly(A), and poly(G), were used for the investigation of AZA–DNA interaction. The electrochemical results were correlated with morphological characterization using FESEM and mass spectrometry investigations.

### 3.1. Azathioprine Redox Behavior

#### 3.1.1. Voltammetric Analysis

Cyclic voltammograms taken at the GC electrode at a scan rate of 100 mVs^−1^ were recorded in 0.1 M pH 4.5 acetate buffer containing 500 µM AZA between the negative potential limit *E*_min_ = −1.0 V and the positive potential limit *E*_max_ = +1.40 V.

When scanning toward positive potential values, on the first positive-going scan of the first voltammogram (Figure 1A), no oxidation reaction appeared, showing that AZA is not electroactive at the experimental conditions. When reversing the scan direction, on the negative-going component of the first voltammogram, a reduction peak 1_c_ appeared at *E*_pc_ = −0.62 V. Subsequently, on the second voltammogram recorded in the same solution and without cleaning the electrode surface, two new anodic peaks, 2_a_ and 3_a_, appeared at *E*_pa_ = +0.45 V and *E*_pa_ = +1.35 V, respectively, corresponding to the electrochemical oxidation of the AZA reduction product. Additionally, a new cathodic charge transfer reaction, 2_c_, occurring at *E*_pc_ = −0.25 V was observed.

A new experiment was performed in similar conditions using a clean GC electrode surface but scanning toward negative potential values (Figure 1B). The cathodic peak 1_c_ appeared at *E*_pc_ = −0.62 V, and after changing the scan direction the two anodic charge transfer reactions, 2_a_ and 3_a_, appeared at *E*_pa_ = +0.45 V and *E*_pa_ = +1.35 V, respectively. Additionally, on the second scan, the reduction peak 2_c_ occurred at *E*_pc_ = −0.25 V. By increasing the pH of the electrolyte (Figure 2), all redox potentials of AZA shifted linearly to less positive potentials, while ΔE_p_ of the 2_a_/2_c_ pair remained constant.

Nevertheless, the effects of dissolved oxygen on the origin of peak 2_c_ were investigated at different pHs using normal oxygen content and O_2_-bubbled solutions (Appendix A). The results revealed that the appearance of the 2_c_ peak was not influenced by the presence of the O_2_ in the electrochemical cell (Appendix A).

The effect of the scan rate on the reduction current of AZA was investigated in a 500 µM AZA solution prepared in phosphate buffer (0.1 M, pH = 7.0; Figure 2). The cyclic voltammograms were recorded for a scan rate of 10 < ν < 500 mVs^−1^. Between measurements, the electrode surface was polished in order to ensure the surface was clean and to avoid possible problems related to the adsorption of AZA or AZA redox products onto the GCE surface. The cyclic voltammograms revealed that the reduction current increased linearly with the increase of the square root of the scan rate and the peak potential shifts toward more negative values, suggesting an irreversible diffusion-controlled process. 

According to the Randles–Sĕvcik equation, the peak current in amperes for a diffusion-controlled irreversible system is given by [33]:*I*_pc_(A) = −0.4463 (*F*^3^/*RT*)^1/2^ × *n* × (*α*_c_*n*′)^1/2^ × *A* × [*c*]_∞_ × *D*_0_^1/2^ × ν^1/2^(1)
where *n* is the number of electrons exchanged in the reduction, *α*_c_ is the transfer coefficient, *A* is the surface area of the electrode (cm^2^), and [*c*]_∞_ is the concentration of the electro-active species in mole cm^−3^. The difference between the peak potential *E*_pc_ and the potential at peak half-height *E*_pc/2_ was ~60 mV. For a diffusion-controlled irreversible system |*E*_pc_- *E*_pc/2_| = 47.7/(*α*_c_*n*’) where *α*_c_ is the cathodic charge transfer coefficient and *n*’ the number of electrons in the rate-determining step, *α*_c_*n*’=0.8. For this calculation, the electroactive surface area of the GC electrode was calculated using the diffusion coefficient of hexacyanoferrate (II) in phosphate buffer at *D*_0_ = 7.35 × 10^−6^ cm^2^ s^−1^ [33]. Using the above equation, the diffusion coefficient for *D*_0_ = 5.20 × 10^−5^ cm^2^ s^−1^ was calculated.

The diffusion coefficient for AZA was also determined using the Wilke–Chang equation [34,35]:*D*_0_ = 7.4 × 10^−8^ × *T* × (*α**_sv_* × *M*_sv_)^−1/2^ × (*η* ×*V*_b,a_^0.6^)^−1^(2)
where *T* is the absolute temperature (K), *α_sv_* is the association coefficient, *M_sv_* is the molecular weight, *η* is the viscosity, and *V*_b,a_ is the molar volume at the normal boiling point. A value of *D*_0_ = 1.22 × 10^−5^ cm^2^ s^−1^ was obtained.

The effect of the pH on the redox behavior of AZA at the GC electrode was investigated using cyclic and DP voltammetry in 100 and 500 µM solutions of AZA in supporting electrolytes with different pH values (Figure 3). The voltammetric results showed that the peak potentials shifted negatively with increasing pH, which suggests that protons participate in the reduction process for azathioprine. For reduction peak 1_c_, the negative shift potential was close to the theoretical value of 59.2 mV per pH unit, meaning that the reduction of AZA involved the same number of electrons and protons; however, although the value of the width at the half-height of the peak was around 75 mV, taking into consideration the previous reports on nitro-derivative compounds, the number of electrons involved was considered to be 2.

Nevertheless, considering the fact that the stock solutions of AZA contained a large amount of DMSO, cyclic voltammograms were recorded in several electrolytes containing DMSO (Appendix A). It was observed that the addition of DMSO did not influence the voltammetric response.

#### 3.1.2. Redox Mechanism

The results presented above showed that AZA undergoes reduction in one step, leading to the formation of reduction products, which in turn undergo oxidation at positive potential values. From a structural point of view, AZA is composed of two main moieties: 1-metyl-4-nitro imidazole and 6-mercaptopurine. In order to understand the redox mechanism, cyclic voltammograms for AZA and its moieties were recorded at the GC electrode in acetate buffer at pH 4.5, with potential limits ranging between −1.00 V and +1.40 V and at *v* = 100 mV s^−1^. The results are compared in Figure 4.

Although the 6-mercaptopurine undergoes redox reactions, none of the peaks observed for this compound were identified in the CV recorded for AZA. In fact, the voltammetric behavior of AZA was dominated by that of 1-metyl-4-nitro imidazole. It is proposed that AZA was electrochemically reduced to compound (2), peak 1_c_, through a process involving 2 electrons and 2 protons (Scheme 1), followed by a coupled chemical reaction mechanism involving2 electrons and 2 protons for the conversion of the NO_2_ group to dihydroxylamine, which is unstable and undergoes dehydration to compound (3). In turn, compound (3) undergoes a quasireversible redox reaction, as demonstrated by the presence of a pair of peaks 2_c_ and 2_a_; on the other hand, irreversible oxidation at higher positive potential values leads to compound (5), a reaction responsible for peak 3_a_ of AZA.

In order to confirm the fact that the origin of the peak 2_c_ is not related to peak 3_a_, cyclic voltammograms were obtained at pH 7.0 in 500 μM AZA after the reduction potential of −0.60 V was applied for 30 s. By recording the voltammograms between the potential limits of *E*_0_ = 0, *E*_min_ = 0.70 V, and *E*_max_ = −0.40 V at different scan rates (100, 500, 1000, and 1500 mV s^−1^), the 2_a_–2_c_ redox pair was observed (Appendix A).

### 3.2. Azathioprine–DNA Interaction

#### 3.2.1. Morphological Characterization

Morphological characterization of the dsDNA electrochemical biosensors was performed using FESEM (Figure 5) in order to highlight the modification of the DNA after interaction with AZA.

Figure 5A depicts the highly porous surface of the screen-printed electrode. The image of the surface of the DNA biosensors (Figure 5B) revealed the spread of dsDNA molecules on the surface of the electrode and its complete coverage, which is essential in order to avoid the non-specific adsorption of drug molecules and to maintain the morphology of the screen-printed electrode. After incubation of the DNA with 1 mM AZA over 20 min, the FESEM image (Figure 5C) showed the formation of clusters due to the action of the immunosuppresive drug; thus, AZA accumulated on the surface of the dsDNA-modified electrode and this process led to highly branched microstructures resulting from the smaller interconnected aggregates. The SEM images obtained for DNA biosensor incubated with AZA showed reorganization of the dsDNA molecules on the electrode surface, i.e., a more densely packed structure. Such behavior (i.e., a reorganization of the dsDNA molecules in a more densely packed structure) on the electrode surface was also observed for dsDNA after interaction with other molecules [20].

#### 3.2.2. DNA Electrochemical Biosensor

To evaluate the ability of AZA to cause conformational changes, e.g., hydrogen bonding cleavage, dsDNA base adducts, or oxidative damage, the AZA–dsDNA interaction was investigated using DP voltammetry. The changes in the dsDNA oxidation peaks for desoxyguanosine (dGuo) and desoxyadenosine (dAdo) in the presence and in the absence of AZA were investigated, along with the occurrence of the oxidation peaks of free guanine (Gua), free adenine (Ade), and the guanine and adenine oxidation products, 8-oxoGua and 2,8-oxoAde.

Initial experiments were performed after incubation of the dsDNA electrochemical biosensor in a solution of 100 μM AZA. After the incubation procedure, the biosensor surface was washed with deionized water in order to remove the loosely-bound AZA molecules and then placed into the electrochemical cell, which contained only buffered electrolyte. DP voltammograms were recorded in both positive and negative potential ranges (Appendix A). The DP voltammogram recorded for the DNA biosensor incubated for 10 min with 100 µM AZA (Appendix A, blue line) showed that the oxidation of the DNA bases occurred with the same oxidation current as for the DNA control biosensor (Appendix A, black line); however, the incubation of the DNA biosensor with AZA under a negative applied potential of −0.60 V (the reduction potential of AZA) resulted in an increase of the oxidation currents of guanosine and adenosine bases (Appendix A, red dotted line). This procedure was chosen for the investigation of the DNA–AZA interaction. After the incubation of the DNA biosensor with 100 µM AZA under applied potential, the biosensor surface was washed with deionized water, transferred in supporting electrolytes at pH 4.5, then DP voltammograms were recorded between *E*_i_ = −0.20 V and *E*_f_ = −0.90 V. The results showed the presence of an AZA reduction peak, with the reduction current decreasing with increasing applied potential time (Appendix A).

In a different experiment, the dsDNA electrochemical biosensor was incubated in a solution of 100 μM AZA at an applied potential of −0.60 V for different durations (Figure 6A). At this potential value, the AZA molecules that diffuse through the solution toward the electrode surface are reduced and the reduction products interact with the immobilized dsDNA layer. It is important to note that each experiment was performed with a new biosensor in order to avoid accumulation of AZA and its reduction products within the immobilized DNA layer.

The voltammograms recorded in these conditions after different incubation times showed increases of the dGuo and dAdo oxidation peaks, in agreement with the unwinding of the dsDNA double helix and exposure of the guanine and adenine residues to the electrode surface. At the same time, at +0.65 V a new oxidation peak appeared, and at high incubation times it overlapped with peak 2_a_, which corresponded to the oxidation of the AZA reduction product.

The next electrochemical experiments were performed and the dsDNA electrochemical biosensors was incubated at an applied potential of −0.60 V for 10 min in solutions of different concentrations of AZA (Figure 6B). In these conditions, both dGuo and dAdo oxidation peaks gradually increased with the AZA concentration, but a higher increase was observed for adenine residues. A similar effect was detected for the peak at +0.65 V.

In order to gain insights into the origin of the peak at +0.65 V, DNA electrochemical biosensors of known primary sequences were used. For this, purine homopolynucleotide single-stranded poly(A) and poly(G) electrochemical biosensors were constructed and tested to assess interactions with AZA (Figure 7).

The DP voltammogram in 0.1 M acetate buffer at pH 4.5 for the poly(A) electrochemical biosensor showed one anodic peak at +1.28 V, corresponding to the oxidation of dAdo (Figure 7A). A newly prepared poly(A) electrochemical biosensor was incubated for 10 min in 500 μM of AZA at an applied potential of −0.60 V. In these conditions, the adenine residue oxidation peak increased but no other addition signal was observed [36].

The DP voltammogram in 0.1 M acetate buffer at pH 4.5 for the poly(G) electrochemical biosensor showed one anodic peak at +1.01 V, corresponding to the oxidation of dGuo (Figure 7B). A newly prepared poly(G) electrochemical biosensor was incubated for 10 min in 500 μM of AZA at an applied potential of −0.60 V. The guanine residue oxidation peak increased and the peak at +0.65 V occurred, corresponding to the formation of 8-oxo-dGuo, which causes oxidative damage to the guanine residues [37].

#### 3.2.3. Mass Spectrometry

Drugs interact with DNA through several mechanisms. Among these, electrostatic binding, intercalation, and adduct formation predominate. Apart from the formation of 8-oxo-dGuo, the experiments described above showed increases of the DNA oxidation peaks, especially in the case of adenine residues (although the contribution of AZA oxidation product 4 at peak 3a was not totally excluded), a characteristic effect of intercalation and adduct formation.

The possibility of adduct formation between DNA bases and AZA electrochemical products was investigated using mass spectrometry. For this investigation, the DNA film was removed from the surface of the biosensor after interaction with AZA at −0.60 V and then subjected to acidic digestion as described in Section 2.3.

The MS spectra of the DNA before interaction with AZA showed signals corresponding to free DNA nucleosides, with m/z values ranging between 225 and 270, with the others being due to free phosphate and sugar groups resulting from the acidic digestion, as indicated in Figure 8A.

The MS spectra of the DNA after interaction with AZA showed signals characteristic of DNA as previously described, while additional peaks corresponding to AZA fragments were observed at 96, 189, 240, and 279 *m*/*z* (Figure 8B). The presence of 8-oxo-dGuo was demonstrated one more time through the signal at the 283 mass-to-charge ratio. Additionally, the MS spectra recorded in these conditions showed new signals at 352 and 362 *m*/*z*, which were attributed to an adduct formation between purine residues and the AZA fragment responsible for the 96 *m*/*z* value.

#### 3.2.4. Interaction Mechanism

The electrochemical experiments with the dsDNA, poly(G), and poly(A) electrochemical biosensors demonstrated conformational modifications within the DNA layer through the increase of both purine residue oxidation peaks after interaction with AZA at −0.60 V. This kind of behavior is typical for an interaction mechanism that involves either intercalation or adduct formation, processes that lead to unwinding of the DNA double helix, exposing the purine basis on the electrode surface, which facilitates their oxidation; however, the results presented above showed that AZA interacts with DNA through a complex mechanism.

The experiments carried out with the dsDNA and poly(G) electrochemical biosensors demonstrated that upon reduction of AZA, 8-oxo-dGuo is formed within the DNA layer, which is a type of oxidative damage and a strong mutagenic compound. In order to explain the formation of the guanine oxidation product, it is proposed that upon applying the conditioning potential of −0.60 V, AZA molecules are reduced and the radical formed during this reaction oxidizes guanine residues, resulting in 8-oxo-dGuo formation (Scheme 2A).

On the other hand, the experiments carried out with the poly(A) electrochemical biosensors showed a different behavior. Since this adenine residue in DNA is oxidized at a higher applied potential than guanine, it is proposed that upon applying the conditioning potential of −0.60 V, AZA molecules were reduced and the radical formed during this reaction was stabilized through the interactions with amino groups of adenine and guanine residues, resulting in the formation of adducts (Scheme 2B). In order to prove this interaction mechanism, MS spectra were recorded, demonstrating the formation of an adduct between purine bases and AZA. In fact, these types of DNA damage by AZA are documented in the literature [38,39], whereby nitropolycyclic aromatic compounds were reduced in situ, reacted with DNA, and subjected to NMR and LC-MS investigations, which demonstrated the formation of adducts with both guanine and adenine residues. Nonetheless, the formation of these types of adducts explains the increase of the DNA oxidation peaks upon interaction with AZA at −0.60 V.

## 4. Conclusions

The redox behavior of AZA was investigated on the GC electrode surface for a wide pH range using cyclic and differential pulse voltammetry, and a redox mechanism was proposed. It was shown that AZA undergoes reduction in one step, leading to the formation of reduction products, which in turn undergo oxidation at positive potential values. Nevertheless, DNA electrochemical biosensors were used for the investigation of the morphological modification of DNA upon the interaction with AZA. Different DNA-based biosensors, such as dsDNA, poly(A), and poly(G), were used for the investigation of the AZA–DNA interaction. The electrochemical results were correlated with morphological characterization using FESEM and mass spectrometry, demonstrating that upon reduction of AZA, 8-oxo-dGuo is formed within the DNA layer, which is a type of oxidative damage and a strong mutagenic compound. A mechanism for the AZA–DNA interaction was proposed.

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
