# Peer review of "Redox Mechanism of Azathioprine and Its Interaction with DNA"

_ijms, 2021, doi:10.3390/ijms22136805_

Round 1

Reviewer 1 Report

The reviewer has found the manuscript interesting and would be glad to see it published, but there are some issues that should be addressed first:

- How the DNA film was removed from the electrode surfaces? Kindly describe the procedure. Also, it is stated (line 470) that “…DNA film was removed…and subjected to acidic digestion as described in the section 3.1”, but there is nothing about the acidic digestion in the section 3.1.

- In fig. 4 the CV for mercaptopurine was taken in anodic direction while the other two compounds were measured by CVs started in cathodic direction? The appearance of the CV beginning indicate so. The CVs for all compounds should be taken in the same way.

- The interpretation of SEM images in fig. 5 rise some questions. How probably such a huge coagulated particles would stay adsorbed on the surface of the electrode? And why the surface layer of dsDNA is not disturbed by the coagulation? In any way, AFM would be much more appropriate method for real observation of spatial conformation of DNA after incubation with any compound.

- It is uneasy to believe that the -600 mV biasing during the incubation have any significant impact. The few nanoampers increase of anodic currents do not sound like a strong evidence. It should be shown more clearly by presenting a graph similar to the one in the figure 6 B with individual curves obtained with biosensors biased with different potentials during the incubation with AZA.

- Figure 6A – the time axis indicates that the second DPV was taken after approximately 12 min incubation, not 5 minutes, as stated in the figure caption.

- What is the reason for the cathodic DPV peak disappearance after 20 min incubation of dsDNA biosensor with AZA? The experiments introduced in the figure S5 should be amended and supported for example with impedance spectroscopy analysis.

- the text contains many typos and errors (examples follows), it should be thoroughly edited.

line 88-89 – novelty relies on or consists of

line 91 – “Different DNA based biosensors, such as dsDNA…” – should be rewritten for example like “Biosensors based on different DNA, such as…”

line 231 – 2a/2c pair instead of par 2a/2c

line 282 – obtained

line 286 – the sentence “The voltammetric results showed with the increasing the pH value.” is not very clear.

line 289 – reduction peak 1a?

The (only) sentence in the first paragraph of the section 3.2.2 should be divided to more sentences to make the paragraph more comprehensive.

line 491 – The electrochemical experiments…

Author Response

The reviewer has found the manuscript interesting and would be glad to see it published, but there are some issues that should be addressed first:

- How the DNA film was removed from the electrode surfaces? Kindly describe the procedure. Also, it is stated (line 470) that “…DNA film was removed…and subjected to acidic digestion as described in the section 3.1”, but there is nothing about the acidic digestion in the section 3.1.

The acidic digestion of DNA before and after incubation with AZA involved removal of the DNA film from the electrode surface. Thus, the DNA film of about 350µg dsDNA (10μL containing 35mg mL-1 dsDNA gel) was removed from the electrode surface and treated with HClO4 9M during 10 min. The digestion was stopped with NaOH 9M after 10 min digestion. All the digested samples of DNA were diluted with purified water and subjected to MS spectrometry. Indeed, the section referred in the above text is 2.3 and not 3.1.

- In fig. 4 the CV for mercaptopurine was taken in anodic direction while the other two compounds were measured by CVs started in cathodic direction? The appearance of the CV beginning indicate so. The CVs for all compounds should be taken in the same way.

The fig. 4 was modified according to the reviewer suggestion.

- The interpretation of SEM images in fig. 5 rise some questions. How probably such a huge coagulated particles would stay adsorbed on the surface of the electrode? And why the surface layer of dsDNA is not disturbed by the coagulation? In any way, AFM would be much more appropriate method for real observation of spatial conformation of DNA after incubation with any compound.

The SEM images obtained for DNA biosensor incubated with AZA showed a reorganization of the dsDNA molecules at the electrode surface, i.e. a more densely packed structure. Such behavior has been also observed for dsDNA after interaction with other molecules, e.g. DNA- methotrexate interaction (Ana Dora Rodrigues Pontinha, Sonia Maria Alves Jorge, Ana-Maria Chiorcea Paquim, Victor Constantin Diculescu and Ana Maria Oliveira-Brett, In situ evaluation of anticancer drug methotrexate–DNA interaction using a DNA-electrochemical biosensor and AFM characterization, Phys. Chem. Chem. Phys., 2011,13, 5227–5234). There is no doubt that AFM is a suitable technique for evaluation of DNA-drug interaction but should be taken into account the fact that Gemini 500 Carl Zeiss Field Emission Scanning Electron Microscope provides ultra-high resolution imaging at low accelerating voltages which allows the investigation of biological samples while avoiding sample polarization issue.

- It is uneasy to believe that the -600 mV biasing during the incubation have any significant impact. The few nanoampers increase of anodic currents do not sound like a strong evidence. It should be shown more clearly by presenting a graph similar to the one in the figure 6 B with individual curves obtained with biosensors biased with different potentials during the incubation with AZA.

As mentioned on the second paragraph of the Section 3.2.2 the initial experiments, involving the DNA biosensor, address to the incubation with AZA and the results showed that the oxidation of the DNA bases occurred with the same oxidation current as in the case of DNA control biosensor. On the other hand, applying the AZA reduction potential, i.e. - 0.6 V, resulted in an increase of the oxidation current of the DNA bases. The increase of the oxidation current of the DNA residues was around 6.5 nA for guanine and 17.5 nA for adenine, which represents an increase of 18 and 11 percent, respectively. Taking into account the high sensibility of the electrochemical methods, especially the differential pulse voltammetry, there is no doubt that this variation should be considered.

- Figure 6A – the time axis indicates that the second DPV was taken after approximately 12 min incubation, not 5 minutes, as stated in the figure caption.

The graph was modified.

- What is the reason for the cathodic DPV peak disappearance after 20 min incubation of dsDNA biosensor with AZA? The experiments introduced in the figure S5 should be amended and supported for example with impedance spectroscopy analysis.

The cathodic peak observed on the Fig. S5 corresponds to the reduction of AZA. As described in the Section 2.3, after the incubation of the DNA biosensor with AZA the electrode was rinsed with purified water in order to remove the weakly bound or unbound molecules and measured in acetate buffer using DPV method. However, applying during the incubation the AZA reduction potential, i.e. -0.6 V, the AZA molecules which interact with DNA film and reach the electrode surface will be reduced. Thus, the reduction current of AZA peak depends on the time for which the potential it was applied; after 20 min all AZA molecules, available at the electrode surface, are reduced during incubation.

- the text contains many typos and errors (examples follows), it should be thoroughly edited.

line 88-89 – novelty relies on or consists of

line 91 – “Different DNA based biosensors, such as dsDNA…” – should be rewritten for example like “Biosensors based on different DNA, such as…”

line 231 – 2a/2c pair instead of par 2a/2c

line 282 – obtained

line 286 – the sentence “The voltammetric results showed with the increasing the pH value.” is not very clear.

line 289 – reduction peak 1a?

The (only) sentence in the first paragraph of the section 3.2.2 should be divided to more sentences to make the paragraph more comprehensive.

line 491 – The electrochemical experiments…

The manuscript text was revised and all the suggestion of the reviewer were taken into account.

Reviewer 2 Report

I thoroughly read this manuscript and found quite interesting results. They have thoroughly investigated the electrochemical behavior of azathioprine (AZA) with deoxyribonucleic acid (DNA) through different types of electrochemical analysis. In my opinion, the results will be useful to understand the electrochemical interaction of AZA with DNA in order to develop a procedure for the safe administration of this drug. I am in favor of the publication of this manuscript in this esteemed journal with a few minor modifications as follows:

  1. Line 142, please include the voltage window at which CV was performed.
  2. I do not see any peak 3a (1.2 V?). I think it is just the background current. Please clarify?
  3. Figure 1 and others, please add the y-axes (current response) for a better understanding of voltammograms. What is the difference between the first and the second scan (in both figures 1 A and B)? What are positive-going and negative-going? Please explain these terms. I do not see any difference between both the figures? Please clarify it.
  4. It will be better if this study will be performed in real samples. It is just a suggestion.

Author Response

I thoroughly read this manuscript and found quite interesting results. They have thoroughly investigated the electrochemical behavior of azathioprine (AZA) with deoxyribonucleic acid (DNA) through different types of electrochemical analysis. In my opinion, the results will be useful to understand the electrochemical interaction of AZA with DNA in order to develop a procedure for the safe administration of this drug. I am in favor of the publication of this manuscript in this esteemed journal with a few minor modifications as follows:

  1. Line 142, please include the voltage window at which CV was performed.

The voltage range used in the voltammetric measurements was Estart= 0V, Emin=-1 V and Emax=1.4 V.

  1. I do not see any peak 3a (1.2 V?). I think it is just the background current. Please clarify?

The oxidation potential of peak 3a is pH dependent occurring at Ep= 1.35 V at pH 4.5 (Figure 1),  Ep=1.20 V at pH 7 (Figures 2 and 3A and Ep=1.00 V) at pH 9.

  1. Figure 1 and others, please add the y-axes (current response) for a better understanding of voltammograms. What is the difference between the first and the second scan (in both figures 1 A and B)? What are positive-going and negative-going? Please explain these terms. I do not see any difference between both the figures? Please clarify it.

All the figures contain the current scale. Generally, on the first scan the redox properties of the examined species (in this case AZA) are investigated while the second scan is dedicated to its redox products electrochemical properties.

Positive-going means scanning the potential from negative to positive values, and in this case the working electrode represents the anode of the electrochemical cell, were the oxidation reactions take place.

Negative-going means scanning the potential from positive to negative values, and in this case the working electrode represents the cathode of the electrochemical cell, were the reduction reactions take place.

  1. It will be better if this study will be performed in real samples. It is just a suggestion.

This may be the subject of another work in which DNA-based biosensors can be used to monitor drugs and drugs metabolites in biological fluids.
